# Emerging Community Pantries in the Philippines during the Pandemic: Hunger, Healing, and Hope

**Alma Espartinez** 

Theology and Philosophy Department, De La Salle-College of St. Benilde, Manila 1004, Metro Manila, Philippines; alma.espartinez@benilde.edu.ph

**Abstract:** This research is a critical approach to the emergence of community pantries during the COVID-19 pandemic as at-once contestatory and transformative narratives, foregrounding the Filipino poor's experience of hunger, suffering, and marginality, while also highlighting their collective hope for a better world. I began by exploring the emergence of the community pantry in the Philippines, which was prompted by the government's inadequate response to the plight of the hungry poor due to prolonged mandatory lockdown in the National Capital Region. I then turned to Emmanuel Levinas' concept of hunger as the basis for the ethical giving displayed in the community pantries, which is a symbolic arena where leadership is questioned and the marginalized voices of the hungry poor are both mainstreamed and articulated. I brought ethical giving into relation with the Jewish concept of Tikkun Olam as the platform for the possibility of healing wounded relations. I constructed a particular weave between the community pantry and the Filipinos' shared experiences of hunger that touches on the ethical that can create liberating spaces for collective hope. In conclusion, I argue that this study is valuable for confronting unexamined assumptions of the relationship between hunger, healing, and hope for critical pedagogy and critical spirituality, which can have significant philosophical and theological implications.

**Keywords:** community pantry; critical spirituality; education; hunger; healing; hope; social justice; Tikkun Olam; Levinas

## 1. Introduction

As bleak as our world appears due to this life-changing pandemic, the acts of kindness and generosity displayed in the community pantry have gone viral, vocal, and visceral. It cuts deep. The nourishment available at the food pantry serves as a temporary relief for the deep hunger Filipino people felt during the pandemic and as a permanent quest for a better life for the hungry poor. The community pantry is a contestatory site, as it is a strong reminder, following Levinas' concept of ethical relation, of our failure to fulfill our commitment to others to display compassionate responsibility and social justice. Food pantries are sites not only of charity but also of activism, advocacy, and food justice (de Souza 2019; Riches and Silvasti 2014; Sastry 2020). In her book, *Feeding the Other*, de Souza (2019) movingly explains that "through this process of listening to the voices of the exploited and oppressed we are changed" (p. 226).

This article highlights the experience of hunger during the COVID-19 pandemic that exposes human negligence, which has led to fractured, fragmented, and wounded relations. It seeks to foreground Filipinos' shared experiences of hunger and how such experiences create spaces for redemptive healing and collective hope amidst the feeling of abandonment, oppression, and marginalization. This study is valuable for confronting unexamined assumptions of the relationship between hunger, healing, and hope for critical pedagogy and critical spirituality, which can have significant philosophical and theological implications. The community pantry provides this narrative: had we not been appallingly remiss in our obligation to feed the hungry, no hungry poor would have sought food in the community pantry.

## 2. Emergence of Community Pantry

The concept behind community pantries is neither new nor unique. Some names used to refer to it are food pantries, food banks, fill-a-bowl, mobile pantry (PR Newswire 2008), or soup kitchen, and all refer to emergency meal programs (DeMaria 2015) to support the hungry poor in times of calamity or pandemic. Whatever name they carry, all have one definite purpose: to address food insecurity (Budd 2021). Even before this pandemic emerged, food pantries already flourished conspicuously in the United States, with some universities assisting students suffering from crippling debt back in 2013. These food banks became all the more visible when the COVID-19 outbreak peaked and millions suddenly found themselves without jobs due to some companies shutting down as dictated by shelter-at-home restrictions (Morello 2021). Some community college students have experienced food insecurity during this pandemic and at some campuses, even adjunct faculty have queued in long lines for food supplies (Budd 2021). Inspired by similar projects such as the Little Free Pantry Movement (MacDonald 2020) and Foodbank that are successful in America and Australia (Coconutsbangkok 2020), Thailand's community pantry project—Too Pan Sook or "Pantries of Sharing"—was launched in early May 2020 (Wattanasukchai 2020). The initiative immediately spread to at least 43 provinces.

Duterte's government has received widespread criticism for its mishandling of the COVID-19 pandemic (Arguelles 2021) and mismanagement of funds. The pandemic relief, called the stimulus package, has not much relieved people's needs during the lockdown in Metro Manila. After Duterte skipped his weekly pandemic public address on 7 April 2021, Filipinos began searching for him amidst the hunger affecting the National Capital Region and other neighboring provinces under the strictest enhanced community quarantine. The hashtag #*nasaanangpangulo* ("Where is the president?") trended on social media, tracking the whereabouts of President Duterte. When Filipinos who were growing hungry by the day needed a leader the most, as the COVID-19 virus was surging forward at an alarming pace, President Duterte was nowhere to be seen not only for a day or two but for two solid, silent weeks (Baizas 2021). In his public address, Duterte forthrightly mentioned that he deliberately refused to make a public appearance while remaining just inside Malacañang the entire time to mock his detractors who wanted him to go public. Poor Filipinos felt they were not worth fighting for and, worse, they had no worth at all. This continued until one woman stood up and said, "I'm tired of complaining. I'm tired of inaction".

On 14 April 2021, Ana Patricia Non, a 26-year-old local furniture entrepreneur, organized a small bamboo cart along Maginhawa Street in Quezon City, Philippines, filled with basic staples, canned goods, fresh produce, facemasks, and other necessities. On cardboard was a handwritten note reading: *Maginhawa Community Pantry: Take what you need. Give what you can*. Non rose to instant fame if similar donation-driven efforts spreading like wildfire around the country is to be any indication. The community pantries essentially target Filipino families who heavily rely on dole-outs as well as the new poor who suddenly find themselves in poverty due to the pandemic. While there is charity food worldwide, the community pantry that spread throughout the Philippines symbolizes so much more. More than just 'sweet charity', to use Janet Poppendieck's words from her book with the same title, the community pantry in the Philippines epitomizes public disillusionment about the government's ineptitude in providing food assistance for the hungry poor during the pandemic (Poppendieck 1999). As it purposely exposed the government's dithering handling of the pandemic situation, the unique flourishing of the community pantries served as a sensitizing call for the government to do all it could, if it has done anything at all, for its starving people. The realization that meaningful and timely assistance would derive from sources other than the government challenges the hegemonic narrative that the government is at the forefront in alleviating hunger among Filipinos. With each bundle of goods handed out, the connection between the community and the people was revitalized and a new collective identity forged. With the rise of community pantries, Filipinos are standing for each other as they wished their national leader had stood for them.



### 3. Localizing Levinas on the Philosophy of Feeding the Hungry

An epidemic of community pantries (Lagman 2021) may well be expressive of what Malcolm Gladwell (2000) describes as a tipping point: 'the moment of critical mass, the threshold, the boiling point' (p. 12). The emergence of community pantries is so inwardly raging yet ostensibly gentle. It does not suggest antagonizing the government but its unmistakable message is crystal clear: people will rely on their own resources.

Community, which is at once definitive and unfolding, continuously seesaws between fits of identity and becoming; it is definitive because it has its boundaries and unfolding because it continually flirts with ambiguity (van Bever Donker 2017). Following Levinas' philosophy of uncertainty, where the future always promises the newness of things, I argue that as the community pantry strengthens the foundation of the community, as a collective of persons with a common goal, it simultaneously dislodges what is to be replaced by what can be. The emergence of the community pantry binds the Filipino people so radically that the first order of the day is a relation with the other that is always an offering of oneself, never an approach with empty hands. This offering brings us to the philosophy of Levinas (1969) that highlights the asymmetrical relation of the self with the other, that is, that any human relationship, to be ethical, is premised on one's prioritizing the other's needs. Levinas' ethical relation focuses on the face-to-face relation, which bears the transition from a movement from "being-for-itself" to "being-for-the-Other" (Levinas and Nemo 1985). Levinas abandons the traditionally accepted I–Other relation, ruled by self-interest and symmetry, and insists that it be governed only by the language of disinterest and asymmetry. The epiphany of the face reminds me that the life of the other matters more than my own (Levinas et al. 1988, p. 172). I am always subordinate to others due to my infinite responsibility to them. While I am in the comforts of my home, some people are forced to be outside, profoundly burdened by this pandemic, looking for the day's meal and not knowing if they will survive the day's hunger. These are the hunger narratives of the homeless, the helpless, and the hopeless.

"Voices of hunger", as de Souza mentioned in *Feeding the Other*, "are powerful, complex, and full of desire". (de Souza 2019, p. 217). The community pantries are grounded in theories of social change, in which an essential requirement is dialogical processes geared toward recovering, centering, and foregrounding the voices of the hungry. The presence of community pantries can move toward the future, which involves consciousness-building and the creation of communicative or participatory spaces in which bodies and voices flow easily (de Souza 2019). This emergency food phenomenon revealed how the narratives of the hungry poor allow for the surfacing of counter-hegemonic discourse and for democratizing the social setting to hear the muted voices of the hungry, the poor, and the oppressed. Marginality exists not only in material relations expressed in the widening gap between the poor and the rich but also in the occlusion of discourses that lie outside the dominant ones that are considered official, authentic, and normal. Knowledge that lies outside the framed and prodded discourse, including stories that center on the lived experiences of the marginalized and oppressed, is obstructed and stigmatized, whereas the logic that shapes the language of domination is privileged, maintained, and sustained. Empowerment for social integration of marginalized groups, such as the elderly, disabled, and poor, coupled with respect for each contribution, would significantly improve the participatory processes. Empowering the poor tilts the balance in the poor's favor and fortifies the ties that bind us.

As a part of the world, the hungry other is considered worldly, with concrete and economic needs. With this being the case, the proper response to the appeal of the other will also have to be concrete, i.e., "economic". The ethical demand proceeding from the face of the other asks very concretely, because the other is a body in need, that I place all my concrete possessions for the service of others. Giving to others the bread from my mouth is offering my very life itself; it is here where suffering makes sense, if it has to make any sense at all. With the epiphany of the other's face, a new orientation was established, namely the ethical orientation, which entails both a disorienting of myself and a reorienting of my existence towards, and directed by, the hungry other. Levinas used the term 'scandal'

to refer to the extreme suffering of the other that is justified for some higher good (White 2012). Echoing St. Augustine, I cannot claim that the suffering of the other—merited or unmerited—results in a greater good, especially when the cause of the other's hunger is my superfluity and repletion. The shame, more than the embarrassment, should haunt me for failing to share amidst my abundance. As Levinas poignantly claimed, "the justification of the neighbor's pain is certainly the source of all immorality"(Levinas 1988, p. 163). The unjustifiable hunger of the indigents and the inexcusable humiliation of the poor are too disgraceful and too absurd to even suggest that something good can come out of it at all. All these sufferings, as Levinas painfully says, are useless (Levinas 1988, p. 159).

"To recognize the other", as Levinas strongly argued, "is to recognize a hunger" (Levinas 1969, p. 75). These powerful words of Levinas remind me what it means to be hungry because I have been hungry myself. Knowing the supreme joy that eating provides elicits the ultimate pain of seeing the hungry other. The same line of thought is echoed by Goldstein when he mentioned that "if we must enjoy food in order for the sacrifice of parting from it to have existential meaning, then we must also know the power of the need for food in order to understand the other's need" (Goldstein 2010, p. 41). If we did not enjoy our own bread, passionately, intensely, even solipsistically, then giving that bread to the other would constitute as nothing more than altruism. To take bread from one's own mouth and give it to the other goes far beyond altruism; it is the manifestation of my original commitment to the good of the other. The new meaning that the community pantry acquires emerges from my absolute obligation to the other, which represents a full acknowledgment of the supreme pain of hunger, my recognition of hunger in the other person, and the willing sacrifice of my own satiety to nourish the other's hunger.

The ethical relation between the community and the self demands to always think of the self with the insistence and persistence of one's unlimited responsibility for the other. More importantly, I argue that it is only through an attempt at thinking of the self in this way that it becomes possible to make sense of the demand of the hungry others to be fed and for the community to be redemptively healed.

## 4. Operationalizing Tikkun Olam for Collective Healing

While many Filipinos languish in hunger, the Duterte government, viewed as not doing enough for the people, is failing miserably in its hunger alleviation, passing on its responsibility to the people themselves and abandoning starving people on their own volition (Suazo 2021; Wong 2021). Being left out by the institutions that have the power to hear the pleas of the hungry but fail to hear well produces the ethical loneliness (Stauffer 2015) of the Filipino hungry that is even more profound and much deeper than simple isolation. This feeling of ethical loneliness splits the oppressed and marginalized from the community; it breaks apart the tie that binds them together. Feelings of desertion amidst the pandemic can make their thoughts collapse. I invite, here, the concept of Tikkun Olam in the healing of severed relationships. Tikkun Olam aims to heal the deeply wounded and broken world, and to transform it into a better and more just society.

Tikkun Olam originates from a Hebrew phrase that means *healing*, rectifying, or mending the world (Berman and Davis-Berman 1999; Rosenthal 2005; Winer 2008) that we may proclaim the beauty of God's creation. The call to do Tikkun Olam is based on the idea of interconnectedness (Berman and Davis-Berman 1999, p. 24), wherein we all share the same humanity and belong to the same God. Seen in this way, Tikkun Olam brings about double healing: as it invites the community's contestatory narratives of being left in destitution, it more significantly creates the healing narrative of the self from the malady of its own selfish interest. This clearly voids any attempt to arrogate onto itself the food assistance given to the other considering that, in essence, what is given by the self to the other is originally *for* and more essentially *from* the other. I need not only give it back to the other; I also need to repent for this crime of stealing in my lifetime. This community pantry initiative uses Tikkun Olam to unite people—with no labeling of givers and takers—to keep their promise of standing by each other. If such unity occurs within the context of a safe, secure setting,

where genuine warmth and understanding can develop, then, for example, the experiences of the Filipino people and the Duterte government can become the experiences of us all. Rather than stressing the hungry Filipino people's suffering and outrage, although this should never be downplayed, this community pantry should strive to complete Tikkun Olam by focusing on mending damaged relationships. Serving the purpose of unity for all can help heal these individuals who begin to see the larger connections. By focusing on the healing relationship, we can complete the work of Tikkun Olam to redress inequity by empowering the subaltern collectives to initiate social transformation for the world's moral repair.

## 5. Destigmatizing the Hungry Poor

The community pantry's objective is to allow the people to take as much as they need and donate whatever they can. As food assistance is given to the hungry poor, it sends a blend of gratitude and shame. The impoverished are thankful for the food support but at the same time are shamed for being labeled as "freeloaders". Garrett Hardin's essay, *The Tragedy of the Commons*, had a straightforward remark on human nature: people think only of their selfish interest for personal advantage and deplete anything they hold in common (Hardin 2009). This self-defeating mindset leads to the tragedy of the commons. "Ruin," Hardin claimed, "is the destination toward which all men rush" (Hardin 2009, p. 246). He further argued that in a world which gives so much premium to the freedom of the commons, this very freedom granted to the commons spells its tragic collapse. Years after '*The Tragedy of the Commons*' was published, Hardin discouraged the provision of food aid to poorer countries, believing that the less able will grow in number at the expense of the more able.

In sharp contrast, Nobel laureate Elinor Ostrom (2000) debunked Hardin's argument, stating that the tragedy of the commons could be circumvented if sustainable collective action is put in place. The features of sustainable collective action, Ostrom argued, include consistent monitoring of the shared resources, adherence to social norms, and good relationships between the community and other multiple levels of authority. Nijhuis (2021), in his *Miracle of the Commons*, concurred with Ostrom's insights when he said that, "far from being profoundly destructive, we humans have deep capacities for sharing resources with generosity and foresight" (Nijhuis 2021, p. 1). In fact, during the pandemic, in the emergence of community pantries, Filipinos proved Hardin's argument wrong. The pantry system was built on trust and with clear guidelines: give according to your means; take according to your needs. To counter this negative view of the poor taking advantage of community pantries, several studies (Clavijo 2020; Hanna 2019; Shepherd et al. 2011) gave contrary opinions concerning the dependency syndrome perpetuated by those sowing malice on charity and volunteerism. Even our very own community pantries are witnesses to the gifts of the poor, which make the food *from* all *for* all. A homeless man took only two oranges from the pantry; despite being asked to take more, the man said: "*This is all I'm going to eat*" (Valenzuela 2021). Two street sweepers each got one head of cabbage which, according to them, was just enough for a sauté for their families (Valenzuela 2021). Most donations come from the rich and the middle class, but ordinary people who are struggling financially are also contributing what little coins or foodstuffs they can share. In seeking out ways to be givers and not just takers, fishers give away their catch while farmers donate baskets of their produce (Wong 2021). This display of fairness and generosity in the community pantry shows that we each have our own "two fishes and five loaves of bread" to share not only from our abundance but even from our necessity. Even in the most challenging times, Ostrom's work can remind us that people are generally altruistic and the future delectably hopeful. Community pantries that sprouted in the Philippines are homegrown ad hoc efforts of some private citizens; they are not meant to stigmatize and demean the poor. What Patricia Non believed was an immediate act of help cascaded into social solidarity that currently spans a nation.

## 6. Critical Spirituality in the Face of Hunger

The word pantry comes from the French term *paneterie*, which means "pain", and the Latin term *panis*, meaning "bread". Its root word says a lot about the nature of the community pantry and what it is used for. Food pantries do more than alleviate hunger by offering bread to the other. They also yield pain and suffering for the giver who needs to share their bread. In Levinas' concept of the epiphany of the face (Levinas 1969), the unwelcome intrusion of someone hungry disturbs my solitude. The hunger of the other unsettles my satiety; it worries, dislocates, and distracts me. In one of the articles I wrote, I expressed how the face of hunger questions my possession: "the face suspends my solipsistic, infantile enjoyment and puts my enjoyment of the world into question. By questioning my possession of the world, the other requires me to establish a distance between myself and my material existence. The face addressing me reveals that my domination has come to an end" (Espartinez 2014, p. 769).

The irruption of the face of the hungry other is the beginning of my understanding of the depth of my commitment to the other. This is also an invocation of critical spirituality that provokes my obligation to the other, which is at once excessive and startling. Gardner defined critical spirituality as that which "gives life meaning, in a way that connects the inner sense of meaning with a sense of something greater" (Gardner 2011, p. 19). There lies the awakening from my apathy, representing a renewal of vision and faith, a rekindling of hope, a restrengthening of courage for a new level of greater shared meaning with others and the universe, as well as a force greater than myself. Furthermore, critical spirituality aims at "seeking meaning in a way that creates wholeness individually and leads to communities that live in sustainable, inclusive and socially just ways" (Gardner 2011, p. 20). Levinas' philosophy of hunger, without a doubt, provokes a critical spirituality that demands a more profound and more significant commitment to the promotion of social justice that remained elusive.

Social justice, however, should not slip into vindictiveness, for it may become as tyrannous as the injustice it seeks to thwart. I insist, here, that our appropriation of Levinas' face-to-face concept offers an ethical framework and a counter-hegemonic discourse that warn us, *"Thou shall not kill"*, or state directly, *"You should not take the food meant for the starving other"*, representing a force of resistance to the tyranny of shaming, oppression, and abandonment. This resistance establishes the counter-narrative expressed through the emergence of community pantries that confronts the structures and relations sustained by the state's indifference to the plight of the poor. The appreciation that comes from being saved rather than shamed, defamed, or deserted becomes an attractive tessellation of possibilities for healing the fragmented, fractured, and flailed relationship. As I witness the hunger and, worse, the death of the hungry and hopeless others, I am essentially and shamefully reminded of my being blameworthy. No justification, no excuse, and only pure guilt. In the presence of this starving other, I am powerless to dismiss the demands issued by blame.

Hunger teaches me to prepare for the other's hunger; it elevates the act of eating to the status of the holy. However, what if I, too, am poor, needy, and homeless like others? Must I still be held responsible? What concrete giving can I offer them when I myself am deprived of what can sustain my daily existence? Levinas insists that we always have something to give, although not simply in terms of economics. Our ethical relation with the other "will always be an offering and a gift" (Levinas 1990, p. 62). The point of Levinas is that our economic condition is immaterial to our responsibility to them. What is important is that we respond, not simply (and not always) in an economic sense, with all that we can give, even if giving runs us dry.

The community pantry organizers and most of the donors knew they had nothing much to offer as they openly admitted during interviews that they, too, are financially impacted by the COVID pandemic crisis (Wong 2021). All they had, similar to the widow's mite, was the simplicity of their life and the sincerity of their heart. Non has only a few mites to her name and those she gave selflessly as her donation to the starving people. Yet,

with all this—or *just this*—she is able to spark the public, making possible the move from people's utter helplessness to collective hope. It is the height of critical spirituality!

## 7. Theorizing a Purely Beyond-the-Self Hope

The task of this section is twofold: first, to explore a new way of conceptualizing hope and, second, to examine the impact of this hope on ethics. As with Levinas' face-to-face relation, such hope is asymmetrical: one hopes for the other without the expectation of reciprocity. Radically dismissing any self-directed hope, Levinas provided a compellingly ethical concept of hope that is purely beyond the self. Building on Levinas' concept of hope, I argue for this kind of hope in the future, which is completely asymmetrical and disinterested. I argue further that this is the kind of hope that retains the dignity and beauty of the I–Other relation, which would otherwise be no more than an economy where the movement goes back to and benefits the self. In truth and in fact, it is challenging to leverage Levinas' notion of hope in hope narratives. The asymmetry and disinterestedness of Levinasian hope make it difficult to understand, mainly because oftentimes, in everyday interactions, when we hope for the best things for others, we prefer they do the same for us. This is reciprocity and it is precisely this that Levinas primordially rejects.

The purely beyond-the-self hope defines our social relationship with and the depth of our commitment to the hungry others. The essential mark of our subjectivity is being chosen and unable to refuse to be responsible to others, thus the free choice not to feed the hungry is no longer possible. I never know further what I am asked but I promise to be for the other. Tikkun Olam, as an expression of hope for a better world, is articulated as a gesture of this commitment: "relation with the other will always be an offering and a gift". I am articulating my hope for the future and committing to be responsible in the face of others' supplication and demand. A just hope responsive to the call of ethics has no guarantee of the future and yet it looks forward. It struggles to recreate itself. It is hope open to surprises. It never rests in complacency, never contented, always struggling; never assured of its position, always restless; never arrogant, always unobtrusive; never static, always evolutionary; never regressive, always progressive. This is the beyond hope that a leader perpetuates if service is to be considered authentic and self-sacrificing.

Although my other-directed hope is seen as a donation of my hope for others, I am forever declared guilty for my dereliction of duty. If my responsibility concerns my being able to be responsive, welcoming, and hospitable, then I am also probably guilty for not being on time to feed the destitute others. The future will always leave open the guilty verdict for me. If this ethical responsibility is not ignored or bypassed, it produces a disturbing relation to the other that motivates a sense of obligation for the other that can alter the public sphere and serve as a source of collective hope in a shared redemptive future (Larocco 2018).

Tikkun Olam provides a theological imperative for social action to redress inequity. No other critical pedagogy matters that does not engage educators and students in a critical dialogue that aims at critiquing hegemonic narratives that silence petite narratives of oppression, marginalization, and alienation. We need to rethink an education system that allows learners to question structures of power and oppression that lead to the rectification of social injustice to improve our society and create a better world. When we see ubiquitous hunger, doing nothing is an inexcusable crime.

Let me summarize my overall line of argumentation. In this article, the emerging of the community pantry is a clear articulation of my obligation to respond to the hunger of others. To preserve the ethical work of feeding the hungry and repairing the world, where hunger is alleviated, healing of our woundedness brought by the self's dereliction of duty should be put in action. In other words, I already carry the hungry other within my very being. There's no way of passing the buck; the buck stops here. I am forever disturbed, displaced, accused, and emptied. I need to guard myself of any irruption of selfish motives that deprive the hungry others of the food that is due to them. I argue further that if our glory for sharing tastes better than the pain of Filipinos dying of hunger, then our motive

betrays our act. Relegating the victims to a position of perpetual marginality is my greatest shame in the face of the hungry others.

## 8. Conclusions

The emergence of community pantries all over the country in April 2021 is the Filipino way of showing that our concern for each other is greater than our desire for divisiveness and vindictiveness. Community pantries in the Philippines serve as a radical expression of human shame derived from the failure to recognize the other's hunger as our own, that the other's insufficiency is our indulgence, and their destitution our negligence. As the moral site of repair and hope, the community pantry is an emblematic field where collective sentiments are articulated and collective wounds both healed and transformed.

A critical spirituality provides, to use George Kent's words, "freedom from want" (Kent 2005) that recognizes the human right to sufficient food and each one's duty to fulfill it. The gradual disappearance of the community pantry measures its success not only by not having people to line up for food with empty stomachs but also by seeing to it that there would be enough food in their very own pantry. This responsibility to others indeed permits, or rather necessitates, us to give the food from our own mouth and offer it to the hungry others. This is the height of the demand of responsibility: I must offer myself to the hungry others before whom, to whom, and for whom I must answer. Our educational institutions cannot teach anything less than this to make their pedagogy responsive to social justice. Constantly, we are left guilty before the poor one who is hungry and in fear of death. Our debt is released only by becoming more responsible. Such are the hyperbolic demands of an ethical philosophy that holds sacred the teaching of being made in the likeness and image of an infinite God. In these gestures of human response, we feel the presence of God passing by. *"Only the man who has recognized the hidden God"*, Levinas emphasizes, *"can demand that He show Himself"* (Levinas 1990, p. 145).

**Funding:** This research study received no external funding.

**Institutional Review Board Statement:** Not applicable.

**Informed Consent Statement:** Not applicable.

**Data Availability Statement:** All the relevant data are already included in the paper.

**Acknowledgments:** I acknowledge the assistance of De Lasalle-College of St. Benilde's Center for Learning Resources for the library materials support it extended to me. I also express my sincerest thanks to the invaluable comments of my peer reviewers; their suggestions made this article clearer.

**Conflicts of Interest:** The author declares no conflict of interest.

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
