# Peer review of "Emerging Community Pantries in the Philippines during the Pandemic: Hunger, Healing, and Hope"

_religions, doi:10.3390/rel12110926_

Round 1

Reviewer 1 Report

Author makes clear connection with contemporary issues in Philippines and Levinas ethics and successfully underlines material and practical aspect in Levinas philosophy.

As I am not familiar with the situation in Philippines not going to make any remarks regarding that.

Few things need some clarification. While describing Levinas ethics, author mentions “ontology of hunger” (14, 288). It might be taken from Goldstein “ontology of eating” but it needs at least some explanation why it is called ontology. 8-To underline the “corporeal ontology” (rather than phenomenology) as does, for example, Tom Sparrow, is possible but author should justify his/her view. Especially having in mind Levinas criticism of ontology.

Author claims that “Levinas gives no indication that my responsibility is in any way limited either by my objective material condition or by that of others who may be better off than I am” (308-310). This is only correct if we deal solely with ethical level. Sure, my responsibility in the face of the other is not limited but limitation comes from the third party. Levinas makes clear that that is never only two of us in the world, there are always the others of the other and so on. The solution is found establishing social structures, that is to say, on political, social level which brings justice for the other and as well for me (and limits my responsibility).

To talk about “mortal sin” (374) in Levinasian context is not accurate.

While quoting or referring it would be useful to refer to exact page or pages.

Author Response

Dear Reviewer:

Thank you so much for your invaluable insights and comments. Please see the attachment. 

Reviewer 2 Report

The paper has contextualized quite well Levinas' philosophy and the Jewish concept of the Tikkun Olam in the social conditions of the Filipinos during the time of the COVID-19 pandemic. It provides a worthy philosophical reading of the Filipino experience with the community pantries, and frames that reading along the lines of solidarity which has always been part of the Filipino psyche and culture. I believe that a review of works about Filipino culture would help demonstrate that 'solidarity' is never alien in that culture as shown in the native concept of 'pakikipagkapwa-tao' (which could roughly be identified with the sense of communion with one's neighbor in the community). 

There is however one point that I'd like to submit for the consideration of the author. The author presents the emergence of the community pantries as an indication also (if not primarily) of failure (of the government [line 12 in the abstract; lines 65 ff.]; and of the people [lines 45-46; line 365]. I however believe that instead of saying that ‘community pantries are indicative of that failure in the personal, social and even political levels’ the community pantries are rather indicative more of the ‘winning’ spirit of the people who realize that ‘even if one’s efforts fail,’ we still have the resources of our communion to rescue us. The community pantry is already a message of hope and healing (despite the tendency of others to put malice into this movement), and highlighting this might be more aligned with the title and problem statement of this paper. If the author could agree with this, I am suggesting that those lines that first blame the government’s failure (something that might need more demonstration too – it may be worthy to note that community pantries are done in bigger scales even in societies like the US which are better able to provide for their people compared to the Philippines – see Jeffrey MacDonald) could be rethought so that the community pantries are better appreciated as a resource especially when things become so dire that even our best personal efforts are not enough. This way, we too could emancipate ourselves from the thought that the ‘government is there as our ultimate provider, and would be the first to blame if we don’t see and experience the worth we think that we deserve as a people.’

Author Response

Dear Reviewer:

This is to express my sincerest thanks for your invaluable comments; your suggestions made this article clearer. Attached are my responses to your comments. Please see the attachment.

Once again, thank you very much. 
